# The Graph Pencil Method:
# Mapping Subgraph Densities to
# Stochastic Block Models

**Lee M. Gunderson**
Gatsby Unit
University College London

**Gecia Bravo-Hermsdorff**
Department of Statistics
University College London

**Peter Orbanz**
Gatsby Unit
University College London

## Abstract

In this work, we describe a method that determines an exact map from a finite set of subgraph densities to the parameters of a stochastic block model (SBM) matching these densities. Given a number $K$ of blocks, the subgraph densities of a finite number of stars and bistars uniquely determines a single element of the class of all degree-separated stochastic block models with $K$ blocks. Our method makes it possible to translate estimates of these subgraph densities into model parameters, and hence to use subgraph densities directly for inference. The computational overhead is negligible; computing the translation map is polynomial in $K$, but independent of the graph size once the subgraph densities are given.

## 1   Introduction

The class of infinitely exchangeable random graphs is a large class of network models that contains all graphon models [6, 21] and stochastic block models (SBMs) [1, 2, 12]. The key statistics of such random graphs are subgraph densities: each fixed, finite graph defines a subgraph density, and it is well known that every exchangeable graph distribution is completely determined by the collection of all its densities of finite subgraphs [12, 21].

In general, an infinite number of densities is required to uniquely determine an infinitely exchangeable graph distribution. For a subclass of models, a finite number of densities suffice. In combinatorics, these random graph distributions are called finitely forcible [21]. In particular, every stochastic block model is completely determined by a finite number of subgraph densities [21]. Such a finite number of subgraph densities can be consistently estimated at a guaranteed asymptotic rate from a single graph-data as this graph grows large [3, 17, 30].

It has long been recognized that the role subgraph densities play for exchangeable network models is analogous to that played by moments for estimation problems on Euclidean sample spaces [5, 7, 12, 22]. On Euclidean space, the method of moments [24, 26] is a powerful inference tool, in particular for mixture models [4, 14]. This suggests that subgraph densities should be well-suited for inference in stochastic block models in particular, since these models are a network analogue of mixture models and only a finite number of densities must be estimated.

Nonetheless, most inference algorithms used in practice (e.g., [8, 9, 13, 18, 25]) do no rely on subgraph densities, but rather on some combination of maximum likelihood estimation, MCMC sampling, variational inference, clustering of the nodes, or other heuristics. The reason is that estimating subgraph densities is only a first step in the inference process — once the densities are known, they must be translated into an estimate of a graphon or stochastic block model.

This paper introduces a new method that does so by generalizing a classical strategy for inferring latent sources (sometimes known as Prony's method [26]) to graph data. The classical Prony's

method is an example of an algorithm that recovers a sparse signal from noisy data, and is related to compressed sensing [28] and the notion of a matrix pencil [23].

Our method proceeds in two steps. The first step (section 3.2) is essentially an application of Prony's method to estimate properties of individual latent blocks, such as their normalized degrees and their relative sizes. The second step (section 4.2) is entirely new; by leveraging the properties inferred in the first step, it uses a generalization of Prony's method to infer properties of *pairs* of the latent blocks, such as the connection properties of a stochastic block model.

The proposed method solves several problems simultaneously: it estimates parameters without laborious and delicate numerical fitting, and makes it possible to efficiently sample from the model without the frequently-encountered problem of degeneracy that plagues many intuitively attractive network models [15, 16, 19]. In contrast to other algorithms for fitting stochastic block models (e.g., [8, 10, 11, 25]) and graphons (e.g, [2, 20, 29]), our method requires essentially zero computational overhead once the subgraph densities have been estimated.

## 2    Notation and Definitions

We use lower-case bold symbols to denote vectors, and upper-case bold symbols to denote matrices and higher-order tensors. Non-bold versions of these symbols refer to particular entries of the bold versions, indicated by their subscripts. We use $\circ$ to denote Hadamard product (i.e., element-wise multiplication), $\bullet$ for inner product, and angled brackets $\langle \, \rangle$ to denote expectation.

A graph $G$ is defined by a set of vertices/nodes $V(G)$ and a set of edges $E(G) \subseteq V(G) \times V(G)$ denoting pairwise connections between nodes. For simplicity, here we focus on undirected, unweighted graphs, with no self-loops or parallel edges. Indexing the $N = |V(G)|$ nodes by the positive integers $[N]$, we can represent $G$ by its $N$-by-$N$ adjacency matrix $\mathbf{A}$, where $A_{ij} = A_{ji} = 1$ if there is an edge between nodes $i$ and $j$, and 0 otherwise.

A stochastic block model $\mathrm{SBM}(\boldsymbol{\pi}, \mathbf{B})$ is defined by: $\boldsymbol{\pi}$, a probability vector of length $K$, which assigns each node to one of $K$ (unobserved) latent blocks/communities; and $\mathbf{B}$, a $K$-by-$K$ connectivity matrix, whose entries $B_{kk'}$ give the probability of an edge connecting a node in block $k$ to a node in block $k'$. As we are considering undirected graphs, $\mathbf{B}$ is symmetric.

Sampling a graph from a $\mathrm{SBM}(\boldsymbol{\pi}, \mathbf{B})$ proceeds in two steps. First, for each node $n \in [N]$, sample its latent block $k(n) \in [K]$ independently from $\boldsymbol{\pi}$. Then, for each pair of nodes $(n, n')$, include an edge between them independently with probability $B_{k(n)k(n')}$. Thus, a stochastic block model defines a distribution over graphs with $N$ nodes for all choices of $N$, and can be identified with the limit of this sequence of distributions as $N \to \infty$.

As mentioned in the introduction, such infinitely exchangeable graph distributions can be characterized by their homomorphism densities[1] $\mu(g)$. Indexed by (some family of) subgraphs $g$, homomorphism densities are the graphical analogue of the classical moments of a distribution [5, 12, 22]. For a stochastic block model, one can compute $\mu(g)$ by summing over all possible assignments of the vertices of $g$ to the blocks of $\mathrm{SBM}(\boldsymbol{\pi}, \mathbf{B})$ [21]:

$$
\underbrace{\mu\big(g\big)\Big|_{\mathrm{SBM}(\boldsymbol{\pi}, \mathbf{B})}}_{\substack{\text{homomorphism density} \\ \mu \text{ of a subgraph } g \text{ in an} \\ \text{SBM given by } \boldsymbol{\pi} \text{ and } \mathbf{B} \\ \text{with } K \text{ blocks}}} = \underbrace{\sum_{\varphi:V(g)\to[K]}^{K^{|V(g)|}}}_{\substack{\text{sum over all maps } \varphi \\ \text{from vertices in } g \text{ to} \\ \text{the } K \text{ blocks}}} \left[ \underbrace{\left( \prod_{i \in V(g)} \pi_{\varphi(i)} \right)}_{\substack{\text{probability of assigning} \\ \text{the vertices to those blocks}}} \times \underbrace{\left( \prod_{(i,j) \in E(g)} B_{\varphi(i)\varphi(j)} \right)}_{\substack{\text{probability of the} \\ \text{corresponding edges}}} \right] \tag{1}
$$

In the following two sections (sections 3 and 4), we describe how to recover the parameters of an SBM from its subgraph densities. Later, in section 5, we describe how to infer these subgraph densities using a graph sampled from such an SBM.

---

[1]Note that the notion of homomorphism only enforces that the edges in $g$ be present; it does not enforce the absence of edges between pairs of nodes in $g$ that do not share an edge. For example, when $g$ has no edges, the homomorphism density is always 1. This is in contrast to the notion of induced subgraph density, which does enforce the absence of those edges.

# 3 Mapping Star Densities to Block Degrees

As the first step of our method, we obtain the normalized degrees of the blocks $\mathbf{d}$ (equation (4)), as well as their relative sizes $\boldsymbol{\pi}$. This can be seen as an application of the classical matrix pencil method to the degree distribution, for which the density of the star subgraphs serve as sufficient statistics.

## 3.1 Classical Coin Collecting

Before we explain our graph pencil method, let us consider the simpler case of a mixture model for infinitely exchangeable sequences of binary variables. Also known as the Bernoulli mixture model [14, 27], such a distribution can be thought of as the outcomes of (some number of) flips of (some number of) biased coins, where each coin is sampled i.i.d. from a (possibly unequal) mixture of $K$ different biases.

To recover the parameters of this model, one must infer the $K$ (unobserved) latent biases $b_k$, as well as the fraction $\pi_k$ of coins with each bias. Note that one does not know which of the coins have the same latent bias (otherwise the inference problem would be trivial).

The moments of this distribution are the expectation of powers of these latent biases, which are indexed by the exponents $r \in \mathbb{N}$:

$$\langle b^r \rangle = \sum_k \pi_k b_k^r \tag{2}$$

Note that these moments can be consistently estimated from the observed data, as we are averaging over the latent variables. From these moments, one can systematically infer the mixture proportion $\boldsymbol{\pi}$ and the biases $\mathbf{b}$ using the matrix pencil method.

In short, construct two matrices $\mathbf{C}$ and $\mathbf{C}'$, with entries $C_{ij} = \langle b^{i+j} \rangle$ and $C'_{ij} = \langle b^{i+j+1} \rangle$:

$$\mathbf{C} = \begin{bmatrix} \langle b^0 \rangle & \langle b^1 \rangle & \cdots & \langle b^{K-1} \rangle \\ \langle b^1 \rangle & \langle b^2 \rangle & \cdots & \langle b^K \rangle \\ \vdots & \vdots & \ddots & \vdots \\ \langle b^{K-1} \rangle & \langle b^K \rangle & \cdots & \langle b^{2K-2} \rangle \end{bmatrix} \qquad \mathbf{C}' = \begin{bmatrix} \langle b^1 \rangle & \langle b^2 \rangle & \cdots & \langle b^K \rangle \\ \langle b^2 \rangle & \langle b^3 \rangle & \cdots & \langle b^{K+1} \rangle \\ \vdots & \vdots & \ddots & \vdots \\ \langle b^K \rangle & \langle b^{K+1} \rangle & \cdots & \langle b^{2K-1} \rangle \end{bmatrix} \tag{3}$$

While not immediately obvious, it is easy to show that the eigenvalues of $\mathbf{C}'\mathbf{C}^{-1}$ are the entries of $\mathbf{b}$. From these, it is straightforward to obtain the associated entries of $\boldsymbol{\pi}$. In the next section, we show why this is the case, while superficially replacing the biases $\mathbf{b}$ of the latent coin types with the normalized average degrees $\mathbf{d}$ of the latent node blocks.

## 3.2 Distilling the Degree Distribution

The first step of our graph pencil method is to apply the standard matrix pencil method to the obtain the average normalized degree of each of the $K$ latent blocks (that is, the likelihood that a random node shares an edge with a node in block $k$). Denoted by $\mathbf{d}$, the entries of this vector are:

$$d_k = \sum_j \pi_j B_{jk} \tag{4}$$

These $d_k$ are latent parameters of the blocks, so we can solve for them in exactly the same way as we did for $b_k$ in the previous coin example. To this end, we consider the analogous moments:

$$\langle d^r \rangle = \sum_k \pi_k d_k^r \tag{5}$$

As before, while $\pi_k$ and $d_k$ are unobserved, the resulting moments can be consistently estimated from observations. In particular, they are precisely the homomorphism densities of the star subgraphs:

$$\langle d^0 \rangle = \mu(\bullet) = 1 \qquad \langle d^1 \rangle = \mu(\mathbf{I}) \qquad \langle d^2 \rangle = \mu(\mathbf{Y}) \qquad \langle d^3 \rangle = \mu(\mathbf{\Psi}) \qquad \ldots \tag{6}$$

We then define the two matrices $\mathbf{C}^\circ$ and $\mathbf{C}^{\mathbf{d}}$, with entries ${C^\circ}_{ij} = \langle d^{i+j} \rangle$ and ${C^{\mathbf{d}}}_{ij} = \langle d^{i+j+1} \rangle$, and the eigenvalues of $\mathbf{C}^{\mathbf{d}}(\mathbf{C}^\circ)^{-1}$ are the normalized degrees of the $K$ blocks. To see why, notice that $\mathbf{C}^\circ$ can be written as a sum of $K$ rank-1 matrices (weighted by $\boldsymbol{\pi}$):

$$
\mathbf{C}^\circ = \sum_k \pi_k
\begin{bmatrix}
d_k^0 & d_k^1 & \cdots & d_k^{K-1} \\
d_k^1 & d_k^2 & \cdots & d_k^K \\
\vdots & \vdots & \ddots & \vdots \\
d_k^{K-1} & d_k^K & \cdots & d_k^{2K-2}
\end{bmatrix}
= \sum_k \pi_k
\begin{bmatrix}
d_k^0 \\
d_k^1 \\
\vdots \\
d_k^{K-1}
\end{bmatrix}
\begin{bmatrix}
d_k^0 & d_k^1 & \cdots & d_k^{K-1}
\end{bmatrix}
$$

$$
=
\begin{bmatrix}
d_1^0 & \cdots & d_K^0 \\
\vdots & \ddots & \vdots \\
d_1^{K-1} & \cdots & d_K^{K-1}
\end{bmatrix}
\begin{bmatrix}
\pi_1 & \cdots & 0 \\
\vdots & \ddots & \vdots \\
0 & \cdots & \pi_K
\end{bmatrix}
\begin{bmatrix}
d_1^0 & \cdots & d_K^{K-1} \\
\vdots & \ddots & \vdots \\
d_K^0 & \cdots & d_K^{K-1}
\end{bmatrix}
$$

$$
= \mathbf{V}\,\mathrm{diag}(\boldsymbol{\pi})\,\mathbf{V}^\top \tag{7}
$$

where we have defined $\mathbf{V}$ as the matrix with entries $V_{jk} = d_k^{j-1}$, i.e.:

$$
\mathbf{V} =
\begin{bmatrix}
d_1^0 & d_2^0 & \cdots & d_K^0 \\
d_1^1 & d_2^1 & \cdots & d_K^1 \\
\vdots & \vdots & \ddots & \vdots \\
d_1^{K-1} & d_2^{K-1} & \cdots & d_K^{K-1}
\end{bmatrix}
\tag{8}
$$

By decomposing $\mathbf{C}^{\mathbf{d}}$ in the same manner, we get:

$$
\mathbf{C}^{\mathbf{d}} = \mathbf{V}\,\mathrm{diag}(\boldsymbol{\pi}\mathbf{d})\,\mathbf{V}^\top \tag{9}
$$

where $\mathrm{diag}(\boldsymbol{\pi}\mathbf{d})$ is the diagonal matrix with entries $\pi_k d_k$.

We now see that $\mathbf{C}^{\mathbf{d}}(\mathbf{C}^\circ)^{-1}$ can be diagonalized as follows:

$$
\begin{aligned}
\mathbf{C}^{\mathbf{d}}(\mathbf{C}^\circ)^{-1} &= \big(\mathbf{V}\,\mathrm{diag}(\boldsymbol{\pi}\mathbf{d})\,\mathbf{V}^\top\big)\big(\mathbf{V}\,\mathrm{diag}(\boldsymbol{\pi})\,\mathbf{V}^\top\big)^{-1} \\
&= \mathbf{V}\,\mathrm{diag}(\boldsymbol{\pi}\mathbf{d})\,\mathbf{V}^\top\big(\mathbf{V}^\top\big)^{-1}\mathrm{diag}(\boldsymbol{\pi}^{-1})\,\mathbf{V}^{-1} \\
&= \mathbf{V}\,\mathrm{diag}(\boldsymbol{\pi}\mathbf{d})\,\mathrm{diag}(\boldsymbol{\pi}^{-1})\,\mathbf{V}^{-1} \\
&= \mathbf{V}\,\mathrm{diag}(\mathbf{d})\,\mathbf{V}^{-1} \tag{10}
\end{aligned}
$$

Hence, the eigenvalues of $\mathbf{C}^{\mathbf{d}}(\mathbf{C}^\circ)^{-1}$ are indeed the normalized degrees $\mathbf{d}$ of the $K$ latent blocks:

$$
\mathrm{eigval}\Big(\mathbf{C}^{\mathbf{d}}(\mathbf{C}^\circ)^{-1}\Big) = \big\{ d_k \big\}_{k \in [K]} \tag{11}
$$

**Remark 1** (**Degree separated assumption**). This is where the assumption of degree-separated blocks is used; if the normalized degrees of the blocks are not unique, then $\mathbf{V}$ will not be invertible.

From the entries of $\mathbf{d}$, we know the entries of $\mathbf{V}$, and can solve a linear system of equations for the corresponding entries of $\boldsymbol{\pi}$:

$$
\sum_k V_{jk} \pi_k = \langle d^{j-1} \rangle \tag{12}
$$

## 4 From Bistar Densities to a Stochastic Block Model

This second step of our graph pencil method is the main insight of our paper. It uses the results from the previous step to reconstruct the connection probabilities $B_{k,k'}$.

## 4.1 Gluing Graphs: An Algebra

We now describe a natural algebra over rooted subgraphs (see Chapter 6 of [21]). We find that using this notation makes our method easier to understand, and facilitates reasoning about its generalizations. For a longer introduction, see appendix A.

Recall the definition of homomorphism subgraph densities of an SBM in equation (1). For a given distribution $\text{SBM}(\boldsymbol{\pi}, \mathbf{B})$, each subgraph $g$ corresponds to a scalar density $\mu(g)$. Rooted homomorphism subgraph densities can be seen as a generalization of this notion. In addition to the subgraph $g$, (singly-)rooted densities also require a designated "rooted" vertex $v \in V(g)$. And instead of a single scalar, they correspond to (a vector of) $K$ scalars $\boldsymbol{\mu}(g, v)$, indexed by the block $k \in [K]$ to which the rooted vertex $v$ is mapped:

$$\underbrace{\mu_k\big(g, v\big)\Big|_{\text{SBM}(\boldsymbol{\pi}, \mathbf{B})}}_{\substack{\text{homomorphism density} \\ \mu \text{ of a subgraph } g \text{ with} \\ \text{rooted vertex } v \text{ in block } k}} = \sum_{\substack{\varphi: V(g) \to [K] \\ \text{s.t. } \varphi(v) = k}}^{K^{|V(g)|-1}} \underbrace{\left[ \left( \prod_{i \in V(g) \setminus v} \pi_{\varphi(i)} \right) \times \left( \prod_{(i,j) \in E(g)} B_{\varphi(i)\varphi(j)} \right) \right]}_{} \quad (13)$$

under: sum over all maps $\varphi$ that send vertex $v$ to block $k$ / probability of assigning the remaining vertices to those blocks / probability of the corresponding edges

The sum is now over maps $\varphi$ that send the rooted vertex $v$ to a particular block $k$, and the probability of assigning the vertices $V(g)$ to blocks includes only the remaining "unrooted" vertices. All the edges $E(g)$ contribute to the product exactly as in equation (1).

A rooted vertex with a single edge corresponds to the normalized degrees of the $K$ blocks:

$$\text{(rooted vertex with an edge)} \quad \mathord{\text{⸭}} \quad \longleftrightarrow \quad \mathbf{d} \quad \text{(normalized degrees of the blocks)} \quad (14)$$

Rooted subgraphs may be combined via the gluing product, denoted by $\circ$. The gluing product corresponds to taking their disjoint union, then merging (i.e., "gluing") the rooted vertices:

$$\text{(gluing rooted vertices)} \quad \mathord{\text{⸭}} \circ \mathord{\text{⸭}} = \mathord{\text{Y}} \quad \longleftrightarrow \quad \mathbf{d} \circ \mathbf{d} = \mathbf{d}^2 \quad \text{(entrywise multiplication)}$$

$$\mathord{\text{⸭}} \circ \mathord{\text{Y}} = \mathord{\text{Y}} \quad \longleftrightarrow \quad \mathbf{d} \circ \mathbf{d}^2 = \mathbf{d}^3$$

$$\mathord{\text{Y}} \circ \mathord{\text{Y}} = \mathord{\text{Y}} \quad \longleftrightarrow \quad \mathbf{d}^2 \circ \mathbf{d}^3 = \mathbf{d}^5 \quad (15)$$

Algebraically, the gluing product corresponds to entrywise multiplication of the vectors, also known as the Hadamard product.

Finally, to convert a rooted subgraph into an observable subgraph density, the rooted vertex may be "unrooted" by taking the dot product with $\boldsymbol{\pi}$:

$$\text{(unrooting vertices)} \quad \mathord{\text{Y}} \quad \longleftrightarrow \quad \mathbf{d}^3 \bullet \boldsymbol{\pi} = \langle d^4 \rangle \quad \text{(dot product with } \boldsymbol{\pi}) \quad (16)$$

Table 1: Operations using homomorphism densities of singly-rooted subgraphs.

| glyph | symbol | meaning |
|---|---|---|
| ⸭ | $\mathbf{d} = \mathbf{B} \bullet \boldsymbol{\pi}$ | vector of the $K$ normalized degrees |
| Y | $\mathbf{d}^2 = \mathbf{d} \circ \mathbf{d}$ | entrywise multiplication |
| Y | $\mathbf{d}^3 \bullet \boldsymbol{\pi}$ | homomorphism density of subgraph |

Before describing how to recover the entries of $\mathbf{B}$, let us summarize the method for recovering the normalized degrees from the previous section using the language of rooted graphs.

For an SBM with $K$ blocks, define a vector $\mathbf{v}$ of (at least) $K$ rooted subgraphs, and take the outer product of this vector with itself where entries are combined via the gluing product. For example:

$$\mathbf{v} = \begin{bmatrix} \circ \\ \mathord{\text{⸭}} \\ \mathord{\text{Y}} \end{bmatrix} \qquad \mathbf{v} \circ \mathbf{v}^\top = \begin{bmatrix} \circ \\ \mathord{\text{⸭}} \\ \mathord{\text{Y}} \end{bmatrix} \circ \begin{bmatrix} \circ \\ \mathord{\text{⸭}} \\ \mathord{\text{Y}} \end{bmatrix}^\top = \begin{bmatrix} \circ & \mathord{\text{⸭}} & \mathord{\text{Y}} \\ \mathord{\text{⸭}} & \mathord{\text{Y}} & \mathord{\text{Y}} \\ \mathord{\text{Y}} & \mathord{\text{Y}} & \mathord{\text{Y}} \end{bmatrix} \quad (17)$$

While the rooted subgraph densities are not directly observable, their unrooted counterparts are, and the entries of $\mathbf{C}^{\circ}$ (equation (3), left) are given by the densities of these unrooted versions:

$$\mathbf{C}^{\circ} = \left(\mathbf{v} \circ \mathbf{v}^{\top}\right) \bullet \boldsymbol{\pi} = \begin{bmatrix} \vdots & \vdots & \vdots \\ \vdots & \vdots & \vdots \\ \vdots & \vdots & \vdots \end{bmatrix} \tag{18}$$

(The glyph containing an unrooted vertex and no edges evaluates to 1.)

To obtain the second matrix $\mathbf{C}^{\mathbf{d}}$ (equation (3), right), glue the rooted subgraph to all entries before unrooting:

$$\mathbf{C}^{\mathbf{d}} = \left(\left(\mathbf{v} \circ \mathbf{v}^{\top}\right) \circ {\scriptstyle\mathbf{\cdot}}\right) \bullet \boldsymbol{\pi} = \begin{bmatrix} \vdots & \vdots & \vdots \\ \vdots & \vdots & \vdots \\ \vdots & \vdots & \vdots \end{bmatrix} \tag{19}$$

Indeed, the reason that the spectrum of $\mathbf{C}^{\circ}(\mathbf{C}^{\mathbf{d}})^{-1}$ gives the normalized degrees $\mathbf{d}$ of the latent blocks is precisely *because* we glued an extra copy of the corresponding rooted subgraph in the construction of $\mathbf{C}^{\mathbf{d}}$. The next section follows a similar recipe.

## 4.2  Extracting the Edge Expectations

The previous matrices $\mathbf{C}^{\circ}$ and $\mathbf{C}^{\mathbf{d}}$ were obtained by taking a dot product that sums over the $K$ blocks. This allowed us to obtain properties of a single block (e.g., its degree $d_k$). The first main insight of this paper is that the matrix pencil method can also be used to obtain properties of pairs of blocks (i.e., their connection probability $B_{kk'}$) by summing over such pairs.

The extension of the gluing algebra to birooted subgraphs is relatively straightforward; a birooted subgraph $(g; u, v)$ has two (distinct) rooted vertices, and their gluing product is given by taking the disjoint union then separately merging the vertices rooted $u$ and the vertices rooted $v$.

Table 2: Operations using homomorphism densities of doubly-rooted subgraphs.

| glyph | symbol | meaning |
|---|---|---|
| | $\mathbf{d}\mathbf{1}^{\top} = \mathbf{L}$ | row matrix of (left) normalized degrees |
| | $\mathbf{1}\mathbf{d}^{\top} = \mathbf{R}$ | column matrix of (right) normalized degrees |
| | $\mathbf{B}$ | matrix of connection probabilities |
| | $\mathbf{L}^2 \circ \mathbf{B} \circ \mathbf{R}$ | entrywise multiplication |
| | $\boldsymbol{\pi} \bullet \left(\mathbf{L}^2 \circ \mathbf{B} \circ \mathbf{R}\right) \bullet \boldsymbol{\pi} = \mu(\quad)$ | observable subgraph density |

For an SBM with $K$ blocks, there are $K + \binom{K}{2}$ degrees of freedom for the entries of $\mathbf{B}$. So we define a vector $\mathbf{v}$ of (at least) $K + \binom{K}{2}$ birooted subgraphs. To this end, we use the symmetric polynomials, with exponent at most $K$, in the two variables and , corresponding to the normalized degrees of the blocks corresponding to the two roots.

$$\mathbf{v} = \begin{bmatrix} \vdots \\ \vdots \\ \vdots \end{bmatrix} \qquad \mathbf{v} \circ \mathbf{v}^{\top} = \begin{bmatrix} \vdots & \vdots & \vdots \\ \vdots & \vdots & \vdots \\ \vdots & \vdots & \vdots \end{bmatrix} \tag{20}$$

In this example for $K = 2$, the second row of $\mathbf{v}$ has been symmetrized with respect to the two rooted vertices, resulting in a formal linear combination of elements in the gluing algebra.

As before, $\mathbf{C}^{\circ\circ}$ is obtained by unrooting the entries, while for the construction of $\mathbf{C}^{\mathbf{B}}$, we glue the rooted subgraph ∘—∘ corresponding to entries of $\mathbf{B}$.

$$\mathbf{C}^{\circ\circ} = \begin{bmatrix} \cdot\cdot & 2\,\text{⸫} & \text{⁝⁝} \\ 2\,\text{⸫} & 2\,\text{⋎}\cdot + 2\,\text{⁝⁝} & 2\,\text{⋎⁝} \\ \text{⁝⁝} & 2\,\text{⋎⁝} & \text{⋎⋏} \end{bmatrix} \tag{21}$$

$$\mathbf{C}^{\mathbf{B}} = \begin{bmatrix} \bullet\!-\!\bullet & 2\,\text{L.} & \text{⊔} \\ 2\,\text{L.} & 2\,\text{⋎.} + 2\,\text{⊔} & 2\,\text{⋎⊔} \\ \text{⊔} & 2\,\text{⋎⊔} & \text{⋎⋏} \end{bmatrix} \tag{22}$$

Note that, once unrooted, isomorphic graphs are equivalent, e.g.: $\text{⋏} = \text{L⸳} = \text{⋎}$

As promised, the eigenvalues of $\mathbf{C}^{\mathbf{B}}(\mathbf{C}^{\circ\circ})^{-1}$ are precisely the entries of $\mathbf{B}$

$$\mathrm{eigval}\!\left(\mathbf{C}^{\mathbf{B}}(\mathbf{C}^{\circ\circ})^{-1}\right) = \left\{ B_{kk'} \right\}_{k \le k'} \tag{23}$$

Moreover, the structure of their corresponding eigenvectors allows us to select specific entries $B_{kk'}$

$$\mathrm{eigvec}\!\left(\mathbf{C}^{\mathbf{B}}(\mathbf{C}^{\circ\circ})^{-1}\right) = \left\{ \begin{bmatrix} 1 \\ d_k + d_{k'} \\ d_k d_{k'} \end{bmatrix} \right\}_{k \le k'} \tag{24}$$

Thus, for each pair of blocks $k$ and $k'$ with normalized degrees $d_k$ and $d_{k'}$, we can estimate their connection probability $B_{kk'}$, estimate the eigenvalue of its corresponding eigenvector:

$$B_{kk'} = \frac{\mathbf{v}^\top (\mathbf{C}^{\mathbf{B}}(\mathbf{C}^{\circ\circ})^{-1})\mathbf{v}}{\mathbf{v}^\top \mathbf{v}} \qquad \mathbf{v} = \begin{bmatrix} 1 \\ d_k + d_{k'} \\ d_k d_{k'} \end{bmatrix}$$

**Remark 2** (**Sufficiency of subgraph densities**). Suppose we fix a finite number $K$ of blocks, and consider all SBMs with $K$ blocks. The $2K - 1$ subgraph densities defined by all stars with up to $2K - 1$ edges then completely determines the degrees and sizes of blocks. Adding the subgraph densities of all bistars with maximum degree at most $K$ then completely determines the connection probabilities between blocks (as a function of their degrees), and therefore the entire distribution of the random graph. In other words, the vector of these densities of stars and bistars is a sufficient statistic for the class of degree-separated stochastic block models with $K$ blocks.

## 4.3 Beyond Degree Correlations: Adding "Two-Hop" Subgraphs

The method described above can be thought of as a "basic" minimum working example of a much more general family of graph pencil methods. The subgraph densities in $\mathbf{C}^{\circ}$ and $\mathbf{C}^{\mathbf{d}}$ are sensitive only to the degree distribution, and the subgraphs in $\mathbf{C}^{\circ\circ}$ and $\mathbf{C}^{\mathbf{B}}$ contain information about the degree-degree correlations. While this is sufficient to recover the parameters of an SBM when the normalized degrees of the blocks are well-separated, the method can be made more robust by adding additional subgraphs.

In particular, define the "two-hop" matrix of connection probabilities $\mathbf{\Lambda} = \mathbf{B} \bullet_{\boldsymbol{\pi}} \mathbf{B}$, where $\bullet_{\boldsymbol{\pi}}$ denotes the matrix product weighted by $\boldsymbol{\pi}$, i.e.:

$$\Lambda_{ij} = \sum_k B_{ik} \pi_k B_{kj} \tag{25}$$

These birooted subgraphs interact the others via gluing and unrooting (see table 3).

Table 3: Incorporating the "two-hop" connection probability $\mathbf{\Lambda}$ into the gluing algebra.

| glyph | symbol | meaning |
|---|---|---|
| ⋀ | $\mathbf{B} \bullet_\pi \mathbf{B} = \mathbf{\Lambda}$ | two-hop connection probability |
| ⋀ | $\mathbf{\Lambda} \circ \mathbf{B}$ | entrywise multiplication |
| ⋀ | $\pi \bullet (\mathbf{\Lambda} \circ \mathbf{B}) \bullet \pi = \mu(\,\triangle\,)$ | homomorphism density of triangles |

We can use this to add more columns to $\mathbf{C^{\circ\circ}}$ and $\mathbf{C^B}$:

$$\mathbf{C^{\circ\circ}} = \begin{bmatrix} \cdots & \cdots & \cdots & \cdots & \cdots & \cdots \\ \cdots & \cdots & \cdots & \cdots & \cdots & \cdots \\ \cdots & \cdots & \cdots & \cdots & \cdots & \cdots \end{bmatrix} \tag{26}$$

$$\mathbf{C^B} = \begin{bmatrix} \cdots & \cdots & \cdots & \cdots & \cdots & \cdots \\ \cdots & \cdots & \cdots & \cdots & \cdots & \cdots \\ \cdots & \cdots & \cdots & \cdots & \cdots & \cdots \end{bmatrix} \tag{27}$$

Remarkably, we can use these larger matrices in exactly the same way as before (using the Moore-Penrose pseudoinverse). As shown in in figure 1, including these additional subgraphs can make the recovery of the SBM more robust.

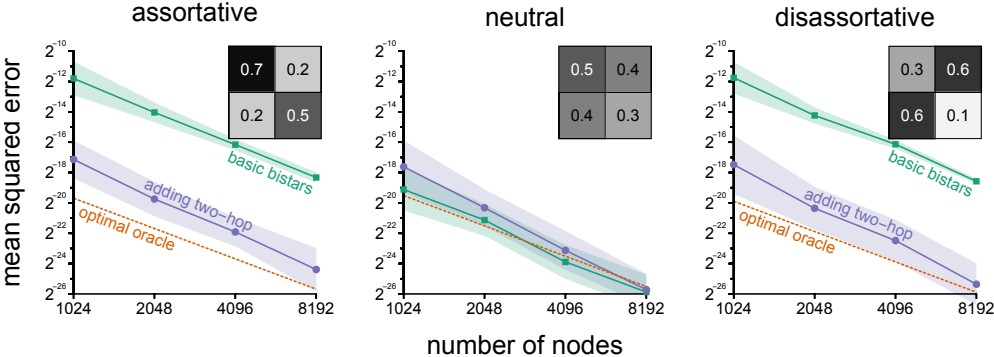

Figure 1: **Adding "two-hop" subgraphs helps recover (dis)assortativity.**
We use 2-by-2 SBMs with varying degrees of assortativity to compare our basic method using bistars from section 4.2 (green squares) and the method that adds the two-hop subgraphs from section 4.3 (purple circles). The vertical axis measures the expected squared error of the probability of an edge between two random nodes, and shading denotes $\pm 1$ standard deviation from the average value. The dashed orange line denotes the expected squared error if the latent blocks of the nodes were known, and both methods appear to converge at this optimal rate. When the SBM is particularly assortative (left) or disassortative (right), the inclusion of two-hop subgraphs results in a notable improvement.

## 5   From an Observed Graph to a Stochastic Block Model

In order to apply the graph pencil method to an observed graph, we need to obtain unbiased estimators of the homomorphism densities used to construct the $\mathbf{C}$ matrices. In particular, for a subgraph $g$ and a (large) graph $G$, we want to compute the fraction of *injective* maps $\varphi$ from $V(g)$ to $V(G)$, such that if $(u, v) \in E(g)$, then $(\varphi(u), \varphi(v)) \in E(G)$.

### 5.1   Quickly Counting (Injective) Homomorphisms

We start with the adjacency matrix $\mathbf{A}$, the identity matrix $\mathbf{I}$, and the all-ones matrix $\mathbf{1}$. Then, we perform our only difficult matrix multiplication: $\sum_k A_{ik} A_{kj} = D_{ij} + \Lambda_{ij}$, where $\mathbf{D}$ is the diagonal

matrix of node degrees, and $\mathbf{\Lambda}$ is the (traceless) "two-hop adjacency" matrix. From these, we can recursively count all the relevant subgraphs using only entrywise multiplication.

We index our subgraphs with a tuple of non-negative integers $(\ell, c, r)$, corresponding to: the number of edges incident to only the "left" node, the number of two-hop paths between the "left" and "right" nodes, and the number of edges incident to only the "right" node $j$. First, we add the two-hop paths:

$$\mathbf{M}^{(0,0,0)} = \mathbf{1} - \mathbf{I} \tag{28}$$

$$\mathbf{M}^{(0,c+1,0)} = \mathbf{M}^{(0,c,0)} \circ \left( \mathbf{\Lambda} - c\mathbf{1} \right) \tag{29}$$

Define the matrix of "left" degrees to be $\mathbf{L} = \mathbf{D} \bullet \left( \mathbf{1} - \mathbf{I} \right) - \mathbf{A}$ (i.e., the degree of node $i$ if node $j \neq i$ were deleted, and zero when $i = j$), and the matrix of "right" degrees to be its transpose $\mathbf{R} = \mathbf{L}^\top$. Next, we add single edges to node $i$, then to node $j$:

$$\mathbf{M}^{(\ell+1,c,0)} = \mathbf{M}^{(\ell,c,0)} \circ \left( \mathbf{L} - (\ell + c)\mathbf{1} \right) \tag{30}$$

$$\mathbf{M}^{(\ell,c,r+1)} = \mathbf{M}^{(\ell,c,r)} \circ \left( \mathbf{R} - (r + c)\mathbf{1} \right) - \ell \mathbf{M}^{(\ell-1,c+1,r-1)} \tag{31}$$

Finally, if an edge connecting the left and right nodes is to be included, we put a mark on the middle integer:

$$\mathbf{M}^{(\ell,c',r)} = \mathbf{M}^{(\ell,c,r)} \circ \mathbf{A} \tag{32}$$

To obtain the counts of a subgraph $g$ in the entire graph, simply sum the entries of the corresponding matrix $\mathbf{M}^{(\ell,c,r)}$. This is analogous to the "unrooting" operation from before.

$$\left[\!\left[ \mathbf{M}^{(\ell,c,r)} \right]\!\right] = \sum_i \sum_j \mathbf{M}^{(\ell,c,r)} \tag{33}$$

For example, $[\![\mathbf{A}]\!]$ is twice the number of edges, as the injective homomorphisms count both orientations. The injective homomorphism densities are obtained by dividing this count by the number of injective maps

$$\mu\!\left( g^{(\ell,c,r)} \right) = \frac{\left[\!\left[ \mathbf{M}^{(\ell,c,r)} \right]\!\right]}{N\left(N-1\right) \cdots \left(N - |V(g)| + 1\right)} \tag{34}$$

## 6 Discussion

The map from a stochastic block model to the density of a given subgraph can be computed analytically (e.g., [12]), or approximated by simulation (sample a graph from the model, count the number of occurrences of the given subgraph, and normalize appropriately). In contrast, the map from the subgraph densities to the stochastic block model is rather nontrivial. This paper provides a way to compute this map.

Since asymptotic rates are known for subgraph densities (see, e.g., [5] and [3]), and the mapping we construct here is differentiable in the interior of its domain, we should expect similar asymptotic rates for recovery of the model parameters. We do not currently know how the rate behaves at the boundaries. At the boundary, the method may in principle output invalid probabilities, as there is at present no explicit constraint enforcing the connection probabilities to be between $0$ and $1$. Hence, as many other methods for inferring SBMs, our method is well-behaved in the interior.

Finally, we note that the method presented here can be extended in a variety of ways, for example to directed edges, weighted edges, directed weighted edges, etc.

## Code

Code implementing our method is available at https://github.com/TheGravLab/TheGraphPencilMethod.

## Acknowledgments and Disclosure of Funding

LMG and PO were funded by the Gatsby Charitable Foundation.

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
