# A The Glyph Algebra

In this paper we use small glyphs as algebraic quantities. This appendix describes how to compute these quantities from the SBM, and the how they relate to the operations used in the inference method.

## A.1 Subgraph Densities

We are interested in the homomorphism density of a subgraph $g$ in a distribution given by $\mathrm{SBM}(\boldsymbol{\pi}, \mathbf{B})$. Consider the probability that 5 randomly sampled nodes form the subgraph $g$ shown below:

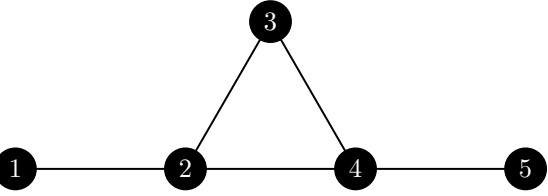

Let $\varphi$ be a map from $V(g)$ to $[K]$ that assigns each node to one of the $K$ blocks. Nodes are sampled independently from the distribution $\boldsymbol{\pi}$, so the probability of such a mapping is given by the product of the corresponding entries of $\boldsymbol{\pi}$:

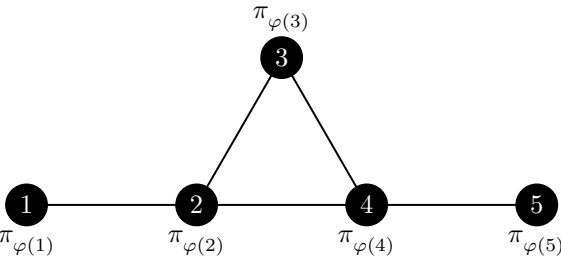

Given one such assignment $\varphi$, the probability of seeing the subgraph $g$ is the product of the edges probabilities, given by the corresponding entries of $\mathbf{B}$:

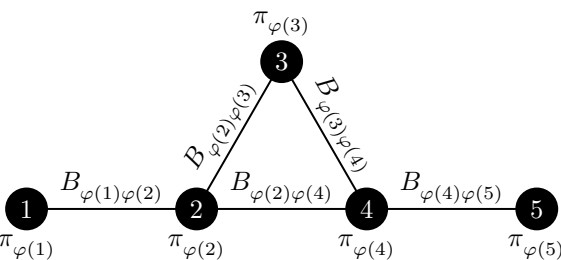

To get the density of subgraph $g$, we sum over all $K^{|V(g)|}$ possible maps:

$$\underbrace{\mu(g)\Big|_{\mathrm{SBM}(\boldsymbol{\pi}, \mathbf{B})}}_{\substack{\text{homomorphism density} \\ \mu \text{ of a subgraph } g \text{ in an} \\ \text{SBM given by } \boldsymbol{\pi} \text{ and } \mathbf{B} \\ \text{with } K \text{ blocks}}} = \underbrace{\sum_{\varphi:V(g)\to[K]}^{K^{|V(g)|}}}_{\substack{\text{sum over all maps } \varphi \\ \text{from vertices in } g \text{ to} \\ \text{the } K \text{ blocks}}} \left[ \underbrace{\left( \prod_{i\in V(g)} \pi_{\varphi(i)} \right)}_{\substack{\text{probability of that} \\ \text{vertex assignment}}} \times \underbrace{\left( \prod_{(i,j)\in E(g)} B_{\varphi(i)\varphi(j)} \right)}_{\substack{\text{probability of the} \\ \text{corresponding edges}}} \right] \tag{35}$$

## A.2 Rooted Subgraph Densities

Consider the "Feynman diagram" for a single edge:

$$\underset{\pi_{\varphi(1)}}{\overset{B_{\varphi(1)\varphi(2)}}{\underset{\textbf{1}}{\bullet}\!-\!-\!\underset{\textbf{2}}{\bullet}}}_{\pi_{\varphi(2)}} \qquad \longleftrightarrow \qquad \sum_{\varphi(1)\in[K]}\sum_{\varphi(2)\in[K]} \pi_{\varphi(1)} B_{\varphi(1)\varphi(2)} \pi_{\varphi(2)}$$

Summing over all node assignments results in a scalar, namely, the edge density of the entire SBM. To represent quantities specific to individual blocks, we consider subgraphs with one of its vertices designated as the "root", which we represent as an open circle.

To compute these quantities, omit the $\sum$ and $\pi$ associated with the rooted vertex:

$$\underset{}{\overset{B_{\varphi(1)\varphi(2)}}{\underset{\textbf{1}}{\circ}\!-\!-\!\underset{\textbf{2}}{\bullet}}}_{\pi_{\varphi(2)}} \qquad \longleftrightarrow \qquad \sum_{\varphi(2)\in[K]} B_{\varphi(1)\varphi(2)} \pi_{\varphi(2)}$$

As we are no longer summing over $\varphi(1) \in [K]$, this computation results in a scalar for each of the $K$ possible assignments of $\varphi(1)$. In other words, the result is a vector of length $K$ (in this case, the vector $\mathbf{d}$ of normalized degrees), indexed by the $K$ blocks.

In general, the length-$K$ vector associated to a rooted subgraph is given by:

$$\underbrace{\mu_k\big(g; v\big)\Big|_{\text{SBM}(\boldsymbol{\pi}, \mathbf{B})}}_{\substack{\text{homomorphism density} \\ \mu \text{ of a subgraph } g \text{ with} \\ \text{rooted vertex } v \text{ in block } k}} = \overbrace{\underbrace{\sum_{\substack{\varphi:V(g)\to[K] \\ \text{s.t. } \varphi(v)=k}}}_{\substack{\text{sum over all maps } \varphi \text{ that} \\ \text{send vertex } v \text{ to block } k}}}^{K^{|V(g)|-1}} \left[ \left( \underbrace{\prod_{i\in V(g)\setminus v} \pi_{\varphi(i)}}_{\substack{\text{probability of that assignment} \\ \text{of remaining vertices}}} \right) \times \left( \underbrace{\prod_{(i,j)\in E(g)} B_{\varphi(i)\varphi(j)}}_{\substack{\text{probability of the} \\ \text{corresponding edges}}} \right) \right] \qquad (36)$$

## A.3 Pointwise Product ≡ Gluing Graphs

Instead of drawing out the large diagrams with labels, we represent the quantities associated with rooted subgraph densities as small glyphs. For example, the vector of normalized degrees is:

$$\text{\large ⦿} \quad \longleftrightarrow \quad \mathbf{d}$$

The graph pencil method uses an operation denoted by $\circ$, which indicates entrywise multiplication (or Hadamard product). In the context of rooted subgraph densities, this is known as the gluing operation [21], as it corresponds to "gluing" together rooted subgraphs by their roots. For example, we can take powers of the vector of degree densities:

$$\text{⦿} \circ \text{⦿} = \text{Y} \quad \longleftrightarrow \quad \mathbf{d} \circ \mathbf{d} = \mathbf{d}^2$$
$$\text{⦿} \circ \text{Y} = \text{Ŷ} \quad \longleftrightarrow \quad \mathbf{d} \circ \mathbf{d}^2 = \mathbf{d}^3$$

The fact that these expressions hold for the glyphs can be seen by factoring the sum over products in equation (36). For example:

$$\text{Y} = \sum_{\varphi(2)\in[K]}\sum_{\varphi(3)\in[K]} B_{\varphi(1)\varphi(2)} B_{\varphi(1)\varphi(3)} \pi_{\varphi(2)} \pi_{\varphi(3)}$$
$$= \left( \sum_{\varphi(2)\in[K]} B_{\varphi(1)\varphi(2)} \pi_{\varphi(2)} \right) \left( \sum_{\varphi(3)\in[K]} B_{\varphi(1)\varphi(3)} \pi_{\varphi(3)} \right)$$
$$= \text{⦿} \circ \text{⦿}$$

## A.4 Inner Product ≡ "Unrooting" Vertices

In section A.2, we saw that when a node is the root, we omit its corresponding $\sum$ and $\pi$ in the expression. In the same way, we can interpret the filling in (or "unrooting") of a vertex as performing

a weighted sum:

$$
\underset{\pi_{\varphi(1)}}{\overset{\text{①}}{\phantom{x}}} \quad \longleftrightarrow \quad \sum_{\varphi(1)\in[K]} \pi_{\varphi(1)}
$$

Thus, given a vector corresponding to a rooted diagram, we can take the inner product, denoted by •, with $\boldsymbol{\pi}$. This corresponds to "unrooting" the rooted vertex, and the result is the corresponding subgraph density for the SBM:

$$
\bullet \, \bullet \, \mathbf{Y} = \mathbf{Y} \quad \longleftrightarrow \quad \boldsymbol{\pi} \bullet \mathbf{d}^3 = \sum_{k=1}^{K} \pi_k d_k^3 = \mu\left(\mathbf{Y}\right)
$$

## A.5 Birooted Subgraph Densities

The procedure for birooted subgraphs is much the same as for singly-rooted subgraphs, except that now there are two designated vertices for which we omit the $\sum$ and $\pi$. This results in a $K$-by-$K$ matrix, indexed by ordered pairs of the $K$ blocks. The simplest example is the birooted edge:

$$
\overset{B_{\varphi(1)\varphi(2)}}{\underset{\text{①}\phantom{xxxxx}\text{②}}{\rule{2cm}{0.4pt}}} \quad \longleftrightarrow \quad B_{\varphi(1)\varphi(2)}
$$

In this case, there is no sum over vertex assignments, and the only term in the product is $B_{\varphi(1)\varphi(2)}$, so this is simply the $K$-by-$K$ matrix of connection probabilities:

$$
\text{o—o} \quad \longleftrightarrow \quad \mathbf{B}
$$

The gluing operation for birooted subgraphs is analogous to that for singly rooted subgraphs, corresponding to entrywise multiplication of $K$-by-$K$ matrices. In glyph notation, each of the rooted vertices in one subgraph is glued to the corresponding rooted vertex in the other. Unrooting the two vertices requires two dot products with $\boldsymbol{\pi}$.

## B  Simulation Details

Our graph pencil method can naturally be extended to use the "two-hop" subgraphs (section 4.3). Figure 1 (reproduced below) shows how doing so can improve the accuracy of the method in some cases (namely, those with notable (dis)assortativity):

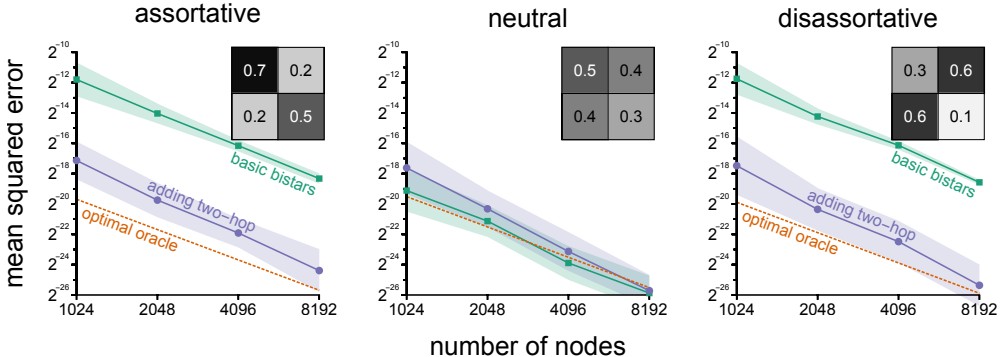

The three figures show the results for three 2-by-2 SBMs with moderate degree separation and varying (dis)assortativity. As all three SBMs have equal-sized blocks $\boldsymbol{\pi} = (0.5, 0.5)$, the grayscaled 2-by-2 matrices displaying the connection probabilities $\mathbf{B}$ are suggestively accurate representations of typical adjacency matrices.

From these SBMs, we sample graphs of different sizes (section 2), compute the necessary subgraph densities (section 5), and infer the parameters $\boldsymbol{\pi}$ and $\mathbf{B}$ (section 4). The green squares represent the

basic method using bistars (section 4.2), and the purple circles represent the method that adds the two-hop subgraphs (section 4.3).

The y-axis measures the mean squared error of the inferred entries of $\mathbf{B}$:

$$\text{squared error} = \boldsymbol{\pi}_{\text{true}}^\top \left( \mathbf{B}_{\text{infer}} - \mathbf{B}_{\text{true}} \right)^2 \boldsymbol{\pi}_{\text{true}} \tag{37}$$

For each SBM, we simulated 16 graphs with 8192 nodes, 32 graphs with 4096 nodes, 64 graphs with 2048 nodes, and 128 graphs with 1024 nodes. Curves show the average squared error for these runs, and shading denotes $\pm 1$ standard deviation, linearized so as to be symmetric on a log plot:

$$\left(\text{mean}, \text{stdev}\right) \longrightarrow \log(\text{mean}) \pm \frac{\text{stdev}}{\text{mean}} \tag{38}$$

The dashed orange line represents the expected squared error if one were to know the latent block assignments of the nodes:

$$\text{Var}\left(B_{ij}\right) = \begin{cases} \frac{B_{ij}\left(1 - B_{ij}\right)}{(n\pi_i)(n\pi_j)} & i \neq j \\ \frac{B_{ii}\left(1 - B_{ii}\right)}{(n\pi_i)^2/2} & i = j \end{cases} \tag{39}$$

And the dashed line shows $\boldsymbol{\pi}_{\text{true}}^\top \text{Var}\left(\mathbf{B}\right) \boldsymbol{\pi}_{\text{true}}$.