# OpenReview forum: "The Graph Pencil Method: Mapping Subgraph Densities to Stochastic Block Models"
_NeurIPS.cc/2023/Conference — NeurIPS 2023 poster_

### Official Review · Reviewer_fpMW · 2023-06-26

**Soundness:** 1 poor
**Presentation:** 2 fair
**Contribution:** 1 poor
**Rating:** 6
**Confidence:** 3

**Summary:**

This paper adapts the matrix pencil method to estimate parameters in models with prescribed subgraph counts, and to simulate from them. A prime example are edge counts and the stochastic blockmodel.

**Strengths:**

The idea to use the matrix pencil method for parameter estimation is interesting and the paper addresses an important problem.

**Weaknesses:**

My main concern is that I do not think that (6) is correct. Focusing on d^3, we have

\langle d^3 \rangle = \sum_k  \pi_k (\sum_j \pi_j B_{jk})^3

Expanding the right hand side gives

 \sum_k  \pi_k \sum_j \pi_j B_{jk} \sum_r B_{rk} \sum_s B_{sk}

However

mu (3-star) = \sum_k \pi_k \sum_{j, r, s distinct} B_{jk} B_{rk} B_{sk}

so the two expressions do not coincide. As equality (6) is the foundation of the proposed approach, I am not convinced of the method.

More generally the paper is not well presented. The abstract mentions exponential random graph models but they do not appear in the main paper at all. A particular version of a stochastic blockmodel is introduced which gives exchangeable edge indicators, and then it is claimed that this can be identified with the limit as the number of vertices tends to infinity. This is not clear at all; in which sense is the limit taken? Two SBMs, one on N vertices and the other one on N+1 vertices, need not be related at all. Is there a coupling construction which maintains exchangeability?

**Questions:**

The homomorphism density does not seem to take care of automorphisms of the counts; is that not an issue?

Is the underlying graph supposed to be finite or infinite? All matrices appear to be finite; why is the notion of a limit important? Are there any theoretical guarantees regarding consistency of the estimation?

There is a lot of literature on graph with prescribed degree distributions, see for example

Britton, T., Deijfen, M., & Martin-Löf, A. (2006). Generating simple random graphs with prescribed degree distribution. Journal of statistical physics, 124, 1377-1397

and

Van Koevering, K., Benson, A. and Kleinberg, J., 2021, April. Random graphs with prescribed k-core sequences: A new null model for network analysis. In Proceedings of the Web Conference 2021 (pp. 367-378).

In Figure 2 in the supplementary material, how many numbers of nodes were used? Was it a step size of 1?

The graphs for Figure 2 are very dense; how does the proposed method work for sparser graphs?


**Limitations:**

There is no mention of any limitation of the method; there is no mention of model mis-specification, and there is no discussion of the variability in the estimates.

---

> ### Author Rebuttal · Authors · 2023-08-10
>
> Thank you for taking the time to review our paper.  The discussions they spurred have been useful for clarifying the paper, as there were some key misunderstandings.
>
> Weaknesses:
> The main concern, about Equation (6), is a misunderstanding by the reviewer, but suggests an excellent way that we can be more clear with our notation.
> When estimating the subgraph densities from a finite graph, the unbased estimators are the \textit{injective} homomorphisms, which require one to sum over distinct nodes.
> However, when computing the subgraph densities from a graphon or Stochastic Block Model, the distinction between injective homomorphism counts (which require that the sums be over distinct labels) and non-necessarily-injective homomorphism counts disappears.  (Essentially this is because as $n\rightarrow\infty$, the fraction of maps that are non-injective goes to zero.)
> So equation (6) is correct, we should be more clear in our notation about whether the quantity is of the graphon/SBM or if it is of a finite graph sample.
>
> Another concern, which was shared by all reviewers, was about the mention of Exponential Random Graph Models in the beginning.  This makes a lot of sense.  We plan to move this to the ending discussion, where the connection to Stochastic Block Models is made (there is some convincing evidence that, for many choices of subgraph constraints, when entropy is maximized, lead to an SBM with some finite number of blocks.).
>
> In terms of identifying the limit of $n\rightarrow\infty$ with a graphon, we can elaborate on this a bit more (also see papers by Chayes and Borgs et al).  Consider a sequence of exchangeable distributions over graphs with $n$ exchangeble nodes, for $n=1,2,\ldots$.  If the each distribution on $n+1$ nodes is related to the distribution on $n$ nodes by randomly deleting a node, then the sequence converges to a graphon (a symmetric measurable function over the unit interval) in terms of the cut norm.
> To make this more clear in the text, we will be explicit that an SBM is just a parameterization of a particular subset of such limiting graphons, and that it does not necessarily have an inherent number of nodes $n$ that must be sampled from it --- it describes all numbers of nodes at once, and therefore the limit as well.
>
> Questions:
> As we are using homomorphism densities, there is no need to worry about automorphism counts in the numerator, so long as you also do not worry about it in the denominator (they cancel out either way).
>
> The "underlying graph" is the underlying graphon/SBM, from which we are sampling graphs of finite size.  As we are estimating the parameters of this model, the only notion of limit is the intuition that "a distribution is the limit of infinite sample size".
>
> The question about sparsity is very good, both theoretically and practically.  We will add another set of experiments in the sparse regime, where we keep the expected degree the same while increasing the number of nodes.

---

> > ### Comment · Reviewer_fpMW · 2023-08-11
> > **Equation (6)**
> >
> > Thank you for your explanations. I am still not satisfied with (6).  Formula (1) details \mu for a SBM on N  nodes. Here N is finite from what I understand; otherwise (1) may be an infinite sum as block sizes will be infinite. So that does not seem to be intended. Consider the simple case of K=1, only one block, that is, an ER graph. Then, with edge probability p = B(1,1) and pi_1 = 1 for all i, and d_1 = p
> > \mu (3-star)  =  4  (n choose 4) p^4. In contrast, < d^3> = p^3. Equation (6) states that these two terms are equal. What am I missing?

---

> > > ### Author Response · Authors · 2023-08-11
> > > **Clearing up Equation (6)**
> > >
> > > Thank you for your prompt reply, it is very nice to have the time to hash this out.
> > >
> > > Equation (1) gives the expression for counting the subgraph density of a subgraph $g$ in an SBM given by $\pi_k$ and $B_{kk'}$ (regardless of the number of nodes $N$ that one decides to sample from the SBM).  It should never be infinite; the $\pi_k$ is a vector of the probability that a random node is in block $k\in [K]$, and the sum is over all $|V(g)|^K$ ways that the $|V(g)|$ nodes in the subgraph could be assigned to each of the $K$ communities in the model.
> > >
> > > So in the case of an ER graph: $K=1$, $\pi_k=(\pi_1)=(1)$, $B_{kk'}=((b_{11}))=((p))$,
> > > then $d_k = (d_1)$, where $d_1 = \sum_k \pi_k b_{kk'} = \pi_1 b_{11} = p$.
> > > And so $\mu(\text{edge}) = \sum_k \pi_k d_k = \pi_1 d_1 = 1 * p = p$,
> > > $\mu(\text{cherry}) = \sum_k \pi_k d_k^2 = \pi_1 d_1^2 = 1 * p^2 = p^2$, and
> > > $\mu(\text{claw}) = \sum_k \pi_k d_k^3 = \pi_1 d_1^3 = 1 * p^3 = p^3$.
> > > These are indeed the homomorphism densities of these subgraphs in an ER graph model on any number of nodes $N$.
> > >
> > > Does this clear it up?  If not, please let me know, it is very helpful in writing the Appendix on Gluing Algebra.
> > >
> > > Very best,
> > > The Authors

---

> > > > ### Comment · Reviewer_fpMW · 2023-08-11
> > > >
> > > > Thank you. Could you please clarify what happens to the first sum in (1), which goes up to | V(g) |   in this simple case?

---

> > > > > ### Author Response · Authors · 2023-08-13
> > > > >
> > > > > Sorry for the delay; thanks for the question.
> > > > >
> > > > > The first sum in (1) is over all maps from the $|V(g)|$ vertices of the subgraph $g$ to the $K$ options for which block each node could be assigned (i.e., the $K$ blocks of the SBM).
> > > > > So for the simple ER case (where there is only $K=1$ block), there is only one map from the vertices of $V(g)$ of any subgraph $g$, as all vertices must map to $k=1$, the only block of the SBM.  Thus, for this simple ER case with $K=1$, the sum in (1) has only a single term, which is the product:
> > > > > $\pi_1 = 1$ (raised to the power of $|V(G)|$, but it doesn't matter), times
> > > > > $b_{11}=p$ (raised to the power of $|E(g)|$),
> > > > > giving $\mu(g)|_{ER_p} = 1^{|V(g)|} * p^{|E(g)|} = p^{|E(g)|}$.
> > > > >
> > > > > As before, please do let us know if you still have questions.
> > > > >
> > > > > Very best,
> > > > > The Authors

---

> > > > > > ### Comment · Reviewer_fpMW · 2023-08-14
> > > > > >
> > > > > > Thank you for the reply. In that case Formula (1) is not correct as it stands. For g having V(g) vertices the sum goes up to | V(g)|  which is more than one term. How would you amend it?

---

> > > > > > > ### Author Response · Authors · 2023-08-14
> > > > > > >
> > > > > > > Oh my gosh, the $|V(g)|^K$ on top of the sum has base and exponent switched --- my sincerest apologies for not seeing it earlier (also thank you for finding a critical typo).
> > > > > > > The correct number of maps $\varphi:V(g)\rightarrow[K]$ is indeed $K^{|V(g)|}$.
> > > > > > > Now the sum for ER ($K=1$) has only $1^{|V(g)|} = 1$ term.
> > > > > > > Equation (13) has the same problem, and both have already been fixed in the new paper.
> > > > > > >
> > > > > > > Very best,
> > > > > > > The Authors

---

> > > > > > ### Comment · Reviewer_fpMW · 2023-08-14
> > > > > >
> > > > > > That makes a lot of sense. I have changed my score to 6.

---

> > > > > > > ### Author Response · Authors · 2023-08-15
> > > > > > >
> > > > > > > Thank you very much.  Not only for the score, but also for your helpfulness and persistence.
> > > > > > >
> > > > > > > Very best,
> > > > > > > The Authors

---

### Official Review · Reviewer_hN5K · 2023-07-06

**Soundness:** 3 good
**Presentation:** 2 fair
**Contribution:** 2 fair
**Rating:** 5
**Confidence:** 3

**Summary:**

The paper studies how to make inference sampling from a stochastic blocking model (SBM) given its corresponding subgraph densities. They first estimate the normalized degrees and the relative sizes of the latent blocks of a SBM, and then infer the connection properties of the SBM using generalized Prony's method.

**Strengths:**

- It is nontrivial study to consider the inverse map from subgraph densities to a stochastic blocking model.
- Overall, the paper is not hard to follow. The proposed method and its properties may have some impacts on its community. As the authors claim, the proposed method can be well generalized to directed, weighted graphs.

**Weaknesses:**

Typos:
L63: entries, give -> entry, gives
L79: explain(ing)
L117: Recall(ing)
L141: allowed -> allows

Suggestions:
Eq.10 looks a bit lengthy, and can be moved to Supplement.

**Questions:**

How to generalize your method to capture hypergraphs with multi-edges between two vertices, signed graphs?
Can you discuss more on how does the method capture large sparse graphs, and graph with structured sparsity?

**Limitations:**

I do not find distinguished limitations of the method. Nonetheless, I would think it could be incremented in the supplement if the method can be easily generalized to directed, weighted graphs.

---

> ### Author Rebuttal · Authors · 2023-08-10
>
> Thank you so much for your time and comments.
>
> Strengths:
> It is nice that you found the paper easy to follow!  We really try to make it easy on the reader, and with the restructuring/moving parts to dedicated Appendices, we think it will be even more accessible.
>
> Weaknesses:
> The typos are much appreciated, and the restructuring makes a lot of sense.
>
> Questions:
> As there are several parts of the main that are being moved to their own dedicated Appendices, there should be some space for us to describe how generalizations to, eg, directed/weighted would look like.
> With respect to multi-edges, there is a mathematically correct way to count weighted subgraph densities in a weighted network (see Lovasz), which when applied to the method described in this paper, would give the "expected number of edges" between a given pair of nodes (instead of the "probability of one edge").
> For directed edges, there are actually more subgraphs one can use (eg, we can reference both the in- and out- degrees of the nodes).
> With respect to sparsity, we will add another set of experiments in the sparse regime, where we keep the expected degree the same while increasing the number of nodes.

---

> ### Author Response · Authors · 2023-08-19
>
> Hello Reviewer,
>
> Thank you again for your thoughtful and detailed review. Please let us know if we have addressed all your concerns.
> In particular, we are including in the main text a section discussing how the method generalizes to directed graphs (with reference to an Appendix for the details).  Is there anything you would like us to elaborate on before the deadline?
>
> Very best,
> The Authors

---

### Official Review · Reviewer_TUi3 · 2023-07-25

**Soundness:** 3 good
**Presentation:** 2 fair
**Contribution:** 3 good
**Rating:** 6
**Confidence:** 4

**Summary:**

Given the subgraph densities of the stochastic block model (SBM), the authors consider the problem of obtaining SBM's parameters (node and edge probabilities). The authors observe a connection between this estimation problem and estimating parameters of an exponential signal. So, they cleverly teleported the classical Prony's method for obtaining signal parameters to obtaining the SBM's parameters. This observation by the authors is the main noteworthy contribution of the paper. Finally, the authors provide analytical formulae for computing the SBM parameters via eigendecompositions of matrices constructed from the subgraph densities of the SBM.

**Strengths:**

The main strength of the paper is to develop a simple and elegant solution for obtaining the parameters of SBM using classical Prony's method. The glue algebra and its connection to extracting edge probabilities seem interesting and intriguing.

**Weaknesses:**

The are three main weaknesses in the paper:
1. it expects a lot of background knowledge from the reader ranging from SBM and its subgraph densities to glue algebra--for e.g., Equations in (1) and (13) appear out of the blue. How are the densities related to SBM?
2. Some of the technical details appear to be heuristic rather than rigorous proof--for e.g., The use of glyphs in matrices and the glue algebra explanation in Sections 3.3 and 3.4 appear more like a heuristic rather than a rigorous proof. Why are the identities in equations (13)-(18) true?
3. The title, the narrative in the abstract and introduction, and the actual problem solved are disconnected. Moreover, the current story of exponential random graphical models (which is not even discussed); titbits on statistical significance (lines 14 -21); and concluding remarks on summary statistics are very broad and unrelated. As an example, the last sentence in the conclusion, “...stepping-stones over two centuries old.” is meaningless for the reader who does not have a background in Prony’s analysis or its historical context.
The paper develops a procedure for obtaining parameters of the SBM model using subgraph densities. I suggest the authors rewrite the introduction/abstract/conclusion directly stating this message.

**Questions:**

1. The authors claim to have generalized Prony's method for their network problem. This generalization needs to be clarified and concrete. The "quoted" generalization in Sections 3.3 and 3.4 seems like using Prony's method for extracting edges. Please justify.
2. It is a well-known fact in probability theory that the moments generally do not specify the distribution completely. In that sense, how can the subgraph densities (analogous to moments) determine the SBM parameters?
3. At the high level, is unlabelling vertices (in Eq (16)) similar to the law of total probability? That is, first consider conditional probabilities (labeled homomorphism densities) and then add them up to get the final probability (unlabelled densities).
4. Eq (34) is a random quantity obtained from a realization drawn from SBM. The authors didn't provide any statistical guarantees on this quantity. For example, how accurate is this quantity compared to the true sub-graph density?
5. I did not completely understand the details in Section 34. In particular, the glue algebra seems hard to understand. Could the authors provide a layman's explanation for extracting the edge expectations?

**Limitations:**

The authors have not discussed the limitations of their current approach. However, I think the proposed method is limited to only SBMs. I suggest the authors to comment on generalizing their method to exponential random graph models (if there is any).

---

> ### Author Rebuttal · Authors · 2023-08-10
>
> Thank you very much for the thoughtful review.
>
> Summary:
> Your summary was super accurate and succinct.  And we very much appreciate the "cleverly teleported".  Your suggestions are really helping the readability our paper, primarily with continuity of the narrative, and an inclusion of a pedagogical "layperson" Appendix on the gluing algebra.
>
> Strengths:
> Thank you!  We also find the gluing algebra "interesting and intriguing", ever since Lovasz used them to prove inequalities between subgraph densities.  And your suggestions are super helpful for making the paper more accessible.
>
> Weaknesses:
> 1)  Equations (1) and (13) are the expressions for the expected injective homomorphism densities of (un/labelled) subgraphs in a graph that was sampled from that SBM.  We will include a citation to Lovasz, but there is also an intuitive building-up of definitions that describes where Equations (1) and (13) come from, and how they dovetail nicely with the gluing algebra.
> 2) Related to the previous point, I'm making an Appendix on Gluing Algebra that is written pedagogically with examples, but makes sure to carefully define the gluing algebra and how to interpret the rooted and unrooted subgraphs.
> It first defines the subgraph densities $\mu_g$ and how they are computed by Equation (1).  Then, to introduce (singly) labeled subgraphs, it shows how the normalized degrees $d_k$ are reflected in the notation as the singly-labelled edge.
> Plugging in these definitions into Equations (13) -- (18), it is relatively straightforward linear algebra to see that and that using them in a matrix this way works (like in Equations (7) -- (11)).
> 3) This makes a lot of sense, we will certainly clean up the narrative.
> One of the original motivations for this work was related to Exponential Random Graph Models, but it no longer really belongs in the abstract/motivation.  It may belong better in the discussion, where we will clarify the connection with the SBMs inferred here (ie, they are often the distribution that is desired when using common choices for ERGMs).
> I rather like the phrase about ``stepping-stones'' at the end, we will add to the Motivation/Background a short paragraph on the historical context of Prony's method and its varied applications.
>
> Questions:
> 1) Sure, maybe "generalization" is not the best word.  We really liked your word: to "teleport" the solution.  Would you mind if we used something along those lines?
> 2) For exchangeable distributions over graphs, a bit more can be said about the structure of the mapping from subgraph densities/moments to distributions with those moments.  If two graphons have the same subgraph densities, then they are related to each other by a measure-preserving transformation of the nodes.  Moreover, we consider stochastic block models here, and these are indeed determined by a finite number of subgraph densities. This fact can be found e.g. in Lovasz’ “Large networks and graph limits”, where it is phrased as all graphons that are step functions (which includes SBMs) being ‘finitely forcible’, see Lovasz’ Corollary 16.47.
> 3) Precisely, and thank you, that is an excellent way to phrase it.  Hopefully, this will be cleared up in the Appendix on Gluing Algebra;
> after showing how to understand labelled nodes as removing the sum over that node, the unlabelling of vertices will naturally be seen as taking the weighted sum of conditional probabilities (ie, multiplying by $\pi_k$ and summing over $k$).
> I promise this Appendix will have many examples for anyone who wants to check their understanding.
> 4) Good point.  There are quite a few works on the statistics of subgraph densities, we will mention the salient ones for dense and sparse graphs.  More generally though, as the section on quickly counting subgraphs, Equations (28) -- (34), addresses the practical issue of quickly counting in a real network the subgraphs required for this method.  It might be more appropriate for it to be moved into another Appendix on Counting Graphs.
> 5) Yes, absolutely, this is an excellent idea.  The Appendix on Gluing Algebra is coming along nicely, and we believe it will help a lot.
>
> Limitations:
> Indeed, there needs to be a section added about the limitations.  For example, while SBMs can approximate any graphon to arbitrary accuracy, there will always be some amount of model misspecification unless the distribution is exactly an SBM.  As for the connection with exponential random graph models, it indeed makes more sense to include them here at the end --- there is some convincing evidence that, for many choices of subgraph constraints, when entropy is maximized, lead to an SBM with some finite number of blocks.  Also important to note would be the difficulty in handling SBMs with entries that are equal or close to 0 and 1.

---

> > ### Comment · Reviewer_TUi3 · 2023-08-19
> > **Increase of the score in the hope that the authors will deliver their promise**
> >
> > Thanks for taking the time to address my questions. The title summarizes what my intention is. As long as the readability of the paper gets improved, the authors can use the phrases or explanations I provided in the review. That being said, I am not yet satisfied with answer to the question on statistics of the random quantity in Eq (34). The authors might think more on this to quantify the uncertainty (in terms of some suitable consistency measures--asymptotic or non-asymptotic) in dealing with a sample sub-graph density. Good luck.

---

> > > ### Author Response · Authors · 2023-08-21
> > >
> > > Thank you very much.  Your comments on improving the flow and readability have already been incredibly helpful, and we will make sure to see them all through.
> > >
> > > In particular, for Equation (34), we are able to cite previous work (e.g. [1] and [2]) on characterizing the distribution of subgraph densities in networks sampled from a graphon/SBM. These previous results on characterizing the distribution are directly applicable will help to provide some notion of how large the sampled network needs to be in order to apply our method for inferring the SBM.
> > >
> > > Thanks again, and very best,
> > > The authors
> > >
> > > References:
> > >
> > > [1] Bickel, P. J., Chen, A. & Levina, E. (2011).  The method of moments and degree distributions for
> > > network models.  *The Annals of Statistics*.  (39) 2280–2301
> > >
> > > [2] Zhang, Y. & Xia, D. (2022).  Edgeworth expansions for network moments. *The Annals of Statistics*. (50) 726-753
> > >
> > > [3] Kaur, G. & Röllin, A. (2021).  Higher-order fluctuations in dense random graph models.  *Electronic Journal of Probability.* (26) 1-26
> > >
> > > Very best,
> > > The Authors

---

> ### Author Response · Authors · 2023-08-19
>
> Hello Reviewer,
>
> Thank you again for your thoughtful and detailed review. Please let us know if we have addressed all your concerns. They have been very useful for streamlining the paper's narriative and writing our Appendix on Gluing Algebras.
> Is there anything more you would like us to elaborate on before the deadline? We are happy to provide further details.
>
> Very best, The Authors

---

### Author Rebuttal · Authors · 2023-08-10

Thank you so much to all the reviewers for taking the time to read our paper.  We really appreciate it being easy and fun to read, so the comments on how to improve the narriative were particuarly appreciated.

The main issues seem to be the mention of Exponential Random Graph Models in the beginning.  This we will move to the end, with motivation for why we are mentioning it.  SBMs (like the ones obtained by our method) are often the distribution that is desired when using common choices for ERGMs.  And there is some convincing evidence that, for many choices of subgraph constraints, when entropy is maximized, the result is an SBM with some finite number of blocks.  We will include this in the discussion.

Also related to structure, we are creating Appendices that give a more detailed introduction to the gluing algebra, how it is used in the method, and related proofs.

Sparsity was also mentioned twice, so we will add another set of simulations where the graphs have constant average degree.

With the space freed up, we can also give examples of further extensions to (for example) weighted and directed graphs.

Thank you so much again, and we are looking forward to hearing back from you.

Very best,
The Authors

---

### Decision · Program_Chairs · 2023-09-21

**Decision:**

Accept (poster)

**Comment:**

This paper developes a graphical generalization of the classical Prony method. The reviewer's agree that the problem is important and the analytical solution is clever. The authors are strongly encouraged to take the suggestions by the reviewers into account as they prepare their final manuscript.